# On the cognitive alignment between humans and machines

**Marco Rothermel**
Department of Psychiatry and Psychotherapy
Philipps University of Marburg
Marburg, Germany
rotherme@students.uni-marburg.de

**Soroush Daftarian**
Department of Psychiatry and Psychotherapy
Philipps University of Marburg
Marburg, Germany
daftaria@students.uni-marburg.de

**Tahmineh Alavikoosha**
Department of Psychiatry and Psychotherapy
Philipps University of Marburg
Marburg, Germany
tahmineh.koosha@uni-marburg.de

**Mohammad-Ali Nikouei Mahani**
Department of Psychiatry and Psychotherapy
Philipps University of Marburg
Marburg, Germany
nikoueim@uni-marburg.de

**Hamidreza Jamalabadi**[*]
Department of Psychiatry and Psychotherapy
Philipps University of Marburg
Marburg, Germany
hamidreza.jamalabadi@uni-marburg.de

## Abstract

In this paper, we explore the psychological relevance, similarity to brain representations, and subject-invariance of latent space representations in generative models. Using fMRI data from four subjects who viewed over 9,000 visual stimuli, we conducted three experiments to investigate this alignment. First, we assessed whether a linear mapping between the latent space of a generative mode, in this case a very deep VAE (VDVAE), and fMRI brain responses could accurately capture cognitive properties, specifically emotional valence, of the visual stimuli presented to both humans and machines. Second, we examined whether perturbing psychologically relevant dimensions in either the generative model or human brain data would produce corresponding cognitive effects in both systems — across models and human subjects. Third, we investigated whether a nonlinear mapping, approximated via a Taylor expansion up to the fifth degree, would outperform linear mapping in aligning cognitive properties. Our findings revealed three key insights: (1) the latent space of the generative model aligns with fMRI brain responses across all subjects tested ($r \approx 0.4$), (2) perturbations in the psychologically relevant dimensions of both the fMRI data and the generative model resulted in highly consistent effects across the aligned systems (both the model and human subjects), and (3) a linear mapping, approximated using Ridge regression, performed as well as or better than all Taylor expansions we tested. Together, these results suggest a universal cognitive alignment between humans and between human-model systems. This universality holds significant potential for advancing our understanding of basic cognitive processes and offers promising new avenues for studying mental disorders.

---

[*]Corresponding author.

# 1 Introduction

Understanding how neural responses in the brain lead to cognitive experiences and how these experiences can be modulated is a central goal in neuro-engineering with promising clinical implications Sani et al. [2021]. For example, individuals with major depressive disorder (MDD) often exhibit a negative cognitive bias, perceiving the emotional valence of visual stimuli more negatively compared to healthy individuals Hamilton and Gotlib [2008].

However, developing effective neuromodulation frameworks to address such biases presents several challenges. First, our understanding of how the brain constructs cognitive experiences remains limited, including the mechanisms by which visual stimuli evoke emotions like pleasantness or arousal Herbet and Duffau [2020], Conwell et al. [2021, 2023]. Second, while neural modulation techniques like transcranial electrical stimulation (tES) can alter brain activity, progress in linking neural changes to specific cognitive outcomes via computational models has been slow Sellers et al. [2024], Camacho-Conde et al. [2023], Hahn et al. [2023a], Teckentrup et al. [2021]. Third, despite clear behavioral and cognitive symptoms in disorders like MDD, we lack a comprehensive understanding of the altered neural patterns that underlie these disorders and the interventions that could reverse their characteristic biases Winter et al. [2024], Hahn et al. [2023b], Oganesian and Shanechi [2024].

A promising avenue to address these challenges involves leveraging the alignment between brain representations and those in artificial intelligence (AI) models Goetschalckx et al. [2021]. If strong alignment exists between the brain's representational spaces and those of AI models, computational systems could help explain how cognition arises from neurobiological components, advancing cognitive computational neuroscience. While many studies have explored neural representational alignment Sucholutsky et al. [2023], focusing mainly on how AI model activations relate to fMRI brain responses, few have examined whether these models naturally represent deeper psychological constructs Goetschalckx et al. [2021], such as emotional valence Conwell et al. [2021].

In this paper, we investigate the cognitive alignment between humans and AI by examining whether the latent space representations of Very Deep Variational Autoencoders (VDVAE) are neurally and psychologically aligned with human brain representations. Specifically, we focus on the following questions: First, can a linear mapping between the latent space of a VDVAE and fMRI brain responses accurately predict emotional valence, a key cognitive property, in response to visual stimuli in both humans and machines? Second, do changes in the latent spaces of the generative model or the human brain produce corresponding effects on cognitive responses in the aligned systems? Finally, we explore whether a nonlinear mapping, approximated using a Taylor expansion up to the fifth degree, can improve the alignment of cognitive properties compared to a linear mapping.

Using fMRI responses from four subjects exposed to over 9,000 visual stimuli Allen et al. [2022], we explore these questions, particularly focusing on emotional valence, given its critical importance in human behavior and mental health.

# 2 Methods and Materials

## 2.1 Dataset

We utilized the publicly available Natural Scenes Dataset (NSD), a comprehensive 7T fMRI dataset Allen et al. [2022]. For our study, we used data from four subjects (sub1, sub2, sub5, sub7) who completed all trials. Our training set comprised a total of 8,859 images and 24,980 fMRI trials (with up to three repetitions for each image), while the test set included 982 images and 2,770 fMRI trials. The test images were consistent across all subjects, whereas the training images varied. We focused on data from the visual cortex, as provided by the NSD General Region-of-Interest (ROI) mask at a 1.8 mm resolution. The ROI consisted of 15,724, 14,278, 13,039, and 12,682 voxels for the four subjects, respectively, encompassing visual areas from the early visual cortex to higher visual areas, including V1 and V2.

## 2.2 Cognitive and Neural Alignment

**Generative Model**: We employed a Very Deep Variational Auto-Encoder (VDVAE) Child [2020], a hierarchical generative model with layers of conditionally dependent latent variables. Each layer of

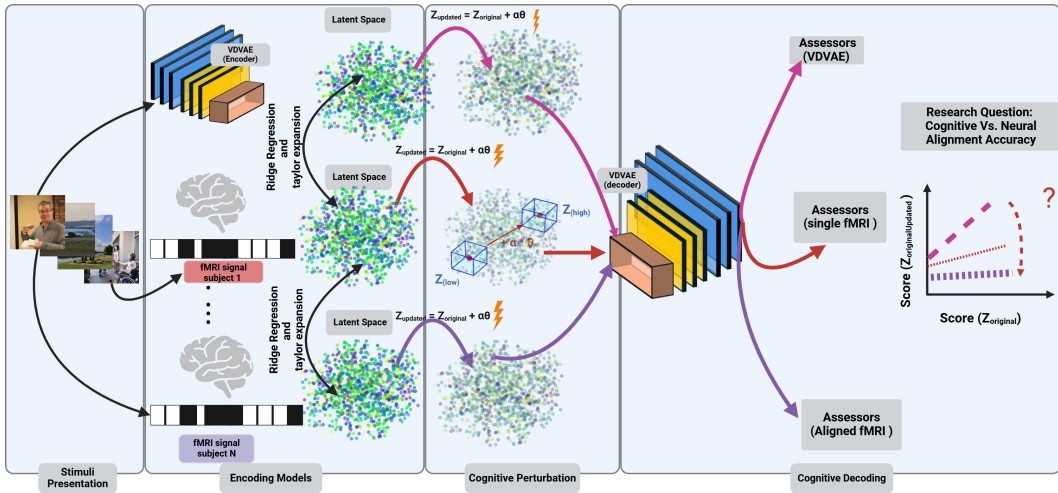

Figure 1: **A Schematic Overview of Cognitive-Neural Alignment and Perturbation Analysis.**
This diagram illustrates the key components of the study, which investigates cognitive alignment
between the VDVAE latent space and fMRI brain activity, and assesses the transferability of cognitive
perturbations across systems. Phase 1: The stimuli presentation phase involves exposing subjects to
a series of images while recording their brain activity using fMRI. The fMRI signals are collected
from individual subjects. Phase 2: The fMRI data and latent space representations from the VDVAE
encoder are used to train encoding models through ridge regression. These models map neural
activity onto latent space representations. In addition to the linear mapping operationalized using
ridge regression, we also employ Taylor expansion to implement nonlinear mapping, testing whether
nonlinear approaches more effectively preserve cognitive properties between humans and machines.
Phase 3: Cognitive perturbations are introduced to the original cognitive scores ($Z_{original}$) by adjusting
the latent space representations, either increasing or decreasing the cognitive scores. Phase 4: The
perturbed cognitive states ($Z_{updated}$) are then decoded back into the original space using the VDVAE
decoder to assess the effect of these perturbations. The cognitive decoding accuracy is evaluated
across three conditions: using the VDVAE, individual fMRI data, and aligned fMRI data across
subjects. The goal is to determine how well cognitive and neural alignment holds under perturbation,
and whether changes in one system (VDVAE or fMRI) lead to corresponding changes in the other.
The study focuses on understanding the relationship between cognitive perturbation and neural
dynamics.

the decoder progressively adds details from coarse to fine, moving from the top to the bottom of the
hierarchy. The top layers capture broad, low-resolution details, while the bottom layers capture finer,
high-resolution features. After training, the VDVAE is capable of generating samples resembling
natural scene images. The original model was trained on a 64×64 resolution ImageNet dataset using
75 layers, resulting in a 91,168-dimensional latent space. However, for our experiments, we utilized
only 31 layers, as no performance changes were observed when including more layers, consistent
with findings from previous studies reconstructing fMRI visual stimuli Ozcelik and VanRullen [2023].

**Neural Alignment**: To align the VDVAE representation with the fMRI data Ozcelik and VanRullen
[2023] and to ensure cross-subject alignment of fMRI data, we used a linear mapping trained via
ridge regression Wehbe et al. [2015], Hoerl and Kennard [1970]. The ridge regression model was
trained exclusively on the training dataset and subsequently tested on a separate, previously unseen
test dataset. This allowed us to map the latent space representations of the VDVAE to the neural
representations observed in the fMRI data and evaluate the alignment across subjects and with the
VDVAE.

**Cognitive Alignment**: To assess the cognitive content, we implemented a two-step procedure. First,
for each visual stimulus, we reconstructed the latent space or fMRI data using the decoder of the
VDVAE. For fMRI data, we first applied ridge regression to map fMRI training patterns to the latent
space of the VDVAE (this is very similar to the first step of the reconstruction technique introduced
by Ozcelik and VanRullen [2023]). The decoder of the VDVAE was then used to generate visual

stimuli from the fMRI brain responses (see Figure 1). Second, we used a pre-trained EmoNet model Goetschalckx et al. [2019] to assess the emotional valence of the reconstructed images, which we assumed served as a proxy for the cognitive information encoded in both the VDVAE latent space and the fMRI data.

## 2.3 Nonlinear Alignment via Taylor Expansion

In addition to testing the linear alignment between the latent space $Z$ of the VDVAE and the fMRI data $X$ using ridge regression, we also explored whether introducing nonlinearities could improve the alignment. Typically, the linear ridge regression model is expressed as $Z = R \cdot X$, where $Z$ represents the latent space of the VDVAE, $X$ represents the fMRI data, and $R$ is the linear ridge regression model that maps $X$ to $Z$.

To account for potential nonlinear relationships between the fMRI data and the VDVAE latent space, we extended the linear model using a Taylor expansion Canuto et al. [2015]. In this case, the latent space $Z$ is expressed as a function of higher-order terms of $X$. Specifically, we expanded the model to include powers of $X$ up to the $n$-th degree, resulting in $Z = [R_1, R_2, \ldots, R_n] \cdot [X, X^2, \ldots, X^n]$, where $R_1, R_2, \ldots, R_n$ represent the regression coefficients for each power of $X$, and $X^2, \ldots, X^n$ are the higher-order terms up to the $n$-th degree. This allows us to capture more complex, nonlinear relationships between the fMRI data and the VDVAE latent space.

For our experiments, we used a Taylor expansion up to the fifth degree (i.e., $n = 5$) to test whether this nonlinear mapping could outperform the linear ridge regression model. We trained this nonlinear model on the training set and evaluated its performance on the test set, comparing the results to the linear model to assess the impact of introducing nonlinearities.

## 2.4 Cognitive Properties as Gradients and Neural-Cognitive Perturbation

Recent work by Goetschalckx et al. [2019] and subsequent studies Younesi and Mohsenzadeh [2022], Goetschalckx et al. [2021] have demonstrated that cognitive properties of images can be linearly modulated along a defined direction within the latent space of generative models. This approach enables the generation of visually similar images that differ in specific cognitive properties. For example, it allows for creating multiple versions of an image (e.g., of a dog) that appear nearly identical but vary in emotional valence. Critically, this cognitively relevant direction is stimulus-independent and can be determined by subtracting the centroids of the latent space representations of images with high versus low scores on the target cognitive property (see Figure 1):

$$\theta_{Z,\text{property}} = \overline{Z}_{\text{high, property}} - \overline{Z}_{\text{low, property}} \tag{1}$$

$$Z_{\text{Updated, Property}} = Z_{\text{org}} + \alpha\theta_{Z,\text{property}} \tag{2}$$

where $\theta_{Z,\text{property}}$ represents the direction in the latent space that modifies the cognitive property, and $\overline{Z}_{\text{high, property}}$ and $\overline{Z}_{\text{low, property}}$ are the mean latent representations of images with high and low cognitive scores, respectively. Applying this direction allows any image to be updated in the latent space to reflect a change in the cognitive property: $Z_{\text{org}}$ represents the original latent representation, $Z_{\text{Updated, Property}}$ represents the modified latent representation, and $\alpha$ is a scaling parameter that can either increase or decrease the property depending on its positive or negative value. Importantly, when a linear relationship exists between the latent spaces of the generative model and fMRI responses, the neural-cognitive gradients ($\theta_{Z,\text{property}}$) in the VDVAE latent space correspond directly to those in the fMRI response space ($\theta_{X,\text{property}}$). This alignment enables the study and comparison of cognitive property effects following perturbations in both humans and machines.

# 3 Results

We first examined whether the cognitive properties, specifically emotional valence, of observed stimuli could be inferred from both the latent space representations of the VDVAE (see Section 2.2) and fMRI brain activity. To quantify this, we compared the emotional valence scores of the original images to those of the reconstructed images across three conditions: (1) using the VDVAE directly, (2) using fMRI data from individual subjects, and (3) using fMRI data from aligned subjects. In

all comparisons, we used the pre-trained assessor, EmoNet Goetschalckx et al. [2019], to infer the emotional valence scores of both the original and reconstructed images.

First, we found that the cognitive properties were most accurately preserved in the VDVAE latent space. The Pearson correlation between the emotional valence scores of the original and reconstructed images in the VDVAE was $r \approx 0.7$ (Figure 2). The less-than-perfect correlation is due to the slightly blurred reconstructions produced by the VDVAE model, as noted also in previous studies Ozcelik and VanRullen [2023]. Next, we decoded the emotional valence scores from the fMRI data of individual subjects. Although the cognitive scores could still be inferred with significant accuracy, there was a notable drop in the Pearson correlation ($r \approx 0.3$), suggesting that cognitive properties were less reliably recovered when using fMRI data from a single subject. This drop points to an imperfect neural-cognitive alignment between the VDVAE latent space and the individual fMRI data. Interestingly, when we aligned the fMRI data across different subjects and then decoded the emotional valence scores from the reconstructed images, the correlation between the decoded cognitive scores across subjects matches the accuracy observed with the VDVAE latent space $r \approx 0.7$. This suggests that the cognitive alignment between subjects becomes nearly perfect once the neural representations are aligned.

Having demonstrated that subjective cognitive experiences can be inferred from both the VDVAE latent space and fMRI brain activity, we next investigated whether this alignment persists under cognitive perturbations (see section 2.4). To test this, we introduced controlled perturbations to the cognitive information in the latent spaces of the VDVAE, the fMRI data of individual subjects, and the aligned fMRI data of all subjects. We then tested whether the changes in cognitive scores were mirrored across the aligned systems. Perturbations were applied in both directions: increasing and decreasing cognitive scores.

Our findings revealed that, across all three conditions — VDVAE, individual fMRI data, and aligned fMRI data — a positive perturbation in the original system produced a positive cognitive change in the aligned systems, while a negative perturbation resulted in a corresponding negative change (see Figure 3). Notably, although the cognitive score changes were directionally consistent across the individual and aligned fMRI data, the magnitude of change was significantly larger in the VDVAE. This difference in effect size suggests two possibilities: either the decoding accuracy of the fMRI data is limited, as we observed previously (see Figure 2), or, more likely, there exists a partial but incomplete alignment between neural and cognitive representations. Specifically, the gradient represented in the fMRI data may differ slightly from that in the VDVAE latent space, indicating that the neural-cognitive mapping captured by fMRI is not fully equivalent to that in the VDVAE model.

# 4 Discussion

In this paper, we investigate the psychological relevance, similarity to brain representations, and subject-invariance of latent space representations in generative models. Our findings offer compelling evidence that this alignment not only exists but also remains stable under cognitive perturbations. However, we observed that while cognitive scores consistently change in the same direction regardless of whether the perturbation is introduced in human or machine data, the effect size of these perturbations depends on both the source (human or machine) and the direction. Specifically, positive changes are more prominent than negative ones, suggesting a nonlinear, potentially compensatory effect in brain dynamics that may contribute to resilience against negative biases Roeckner et al. [2021]. This indicates that although human cognitive representations are aligned with those of the AI model, key differences exist in how information is encoded, particularly at the extremes of cognitive states.

Crucially, we found that aligning neural representations across individuals results in near-perfect cognitive alignment. This finding supports previous studies indicating that neural activations can explain most cognitive variance Conwell et al. [2021], suggesting a robust neural-cognitive alignment in both humans and machines. In other words, aligning fMRI voxel activations across subjects appears to fully align the tested cognitive dimensions, implying that individual differences in cognitive processing may arise directly from variations in neural activation patterns. If this result can be replicated across other cognitive domains, such as control and arousal Horikawa et al. [2020], it could significantly enhance our understanding of human cognition and open new possibilities for AI-driven models to predict and potentially modulate cognitive states.

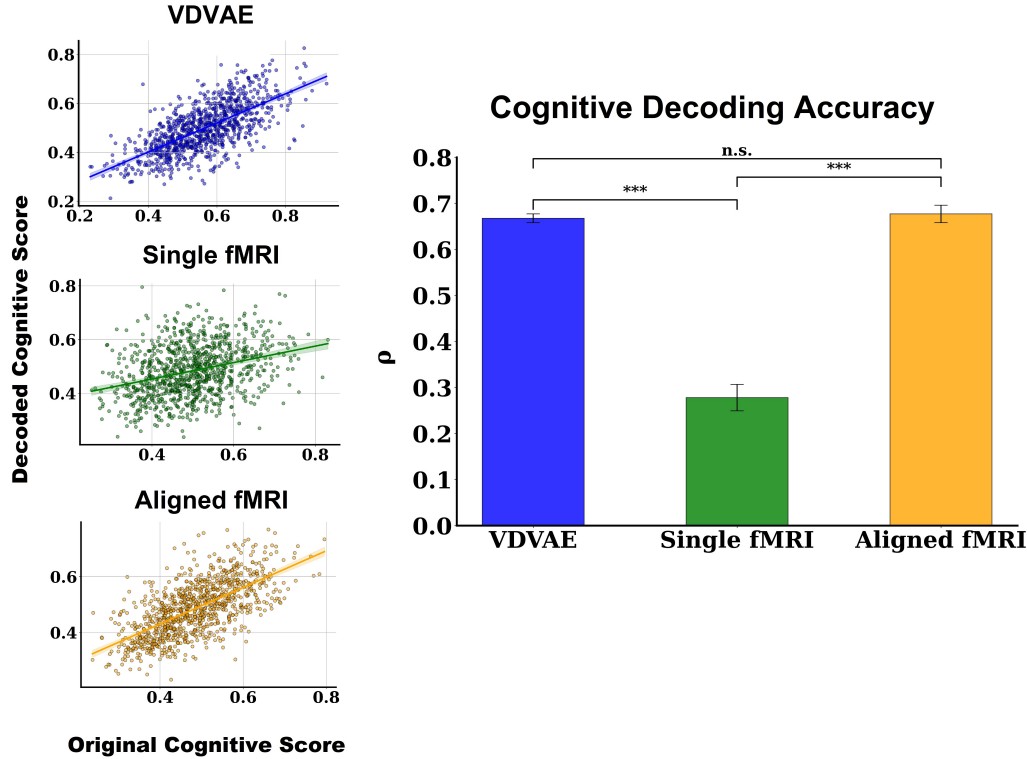

Figure 2: **Cognitive Alignment Between Machine and Humans**. (Left) Scatter plots illustrate the Pearson correlations between the emotional valence scores of original images and the corresponding decoded scores across three conditions: (1) decoding from the latent space of the VDVAE model (blue), (2) decoding from fMRI data of individual subjects (green), and (3) decoding from aligned fMRI data across subjects (orange). The VDVAE achieves the highest accuracy with a Pearson correlation of $r \approx 0.7$, indicating robust preservation of cognitive properties. In contrast, decoding from individual fMRI data shows a notable drop in accuracy, with $r \approx 0.3$, reflecting a weaker alignment between cognitive scores and individual neural data. However, aligning the fMRI data across subjects restores the correlation to $r \approx 0.7$, matching the VDVAE's performance. (Right) The bar plot compares the decoding accuracy ($\phi$) across the three conditions. The VDVAE and aligned fMRI data yield similar high decoding accuracies, significantly outperforming individual fMRI decoding (*** $p < 0.001$, ** $p < 0.01$). These results suggest that cognitive alignment between individuals can be achieved by aligning neural representations, matching the performance of the VDVAE latent space. Nonlinear mappings were also tested but did not show significant improvement over the linear mapping, indicating that linear relationships sufficiently capture cognitive properties in this setup.

Furthermore, our results contribute to the ongoing debate on the manifestation of mental disorders in neural activations. Current machine learning approaches have struggled to decode significant cognitive differences in neural data Winter et al. [2024], prompting some researchers to question whether this goal is even achievable Chekroud et al. [2024]. Our findings suggest a possible explanation: if neural and cognitive data from different subjects are misaligned at the voxel level, accurately decoding cognitive information becomes challenging unless functional alignment is achieved, a step often missing in conventional preprocessing of machine learning studies on mental disorders Walter et al. [2019]. Although not detailed in this paper, our analysis reveals that when neural activity is not aligned across subjects, cognitive score decoding accuracy drops to near zero, consistent with findings in mental health research. These findings underscore voxel-level alignment as potentially essential for reliable decoding of cognitive states across individuals Anderson et al. [2024].

Additionally, these findings highlight the value of linear mappings between brain activity and generative models for understanding cognition. Despite testing nonlinear approaches, linear mappings

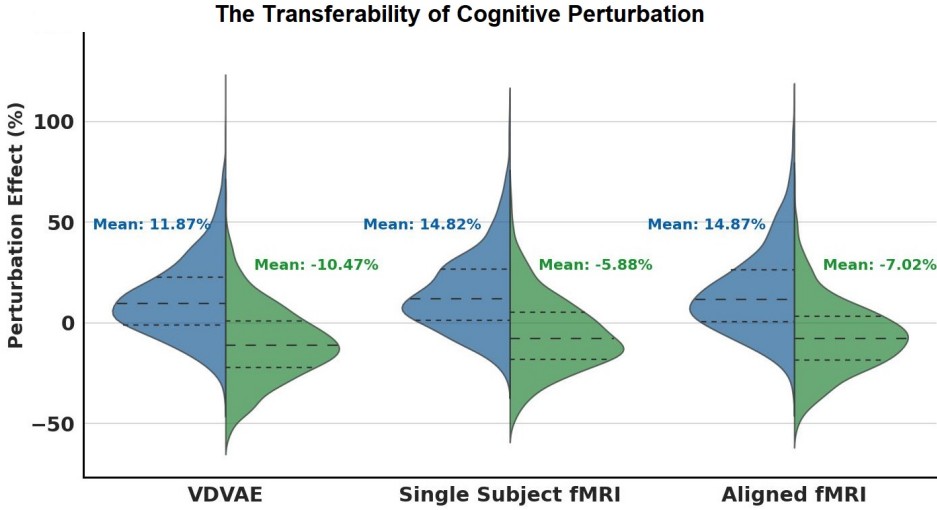

Figure 3: **The Transferability of Cognitive Perturbation Between VDVAE and fMRI Data**. Violin plots show the distribution of perturbation effects on cognitive scores across three conditions: (1) cognitive perturbations applied to the VDVAE latent space, (2) perturbations applied to fMRI data from individual subjects, and (3) perturbations applied to aligned fMRI data across subjects. The perturbation effect represents the percentage change in cognitive scores resulting from both positive and negative perturbations.

consistently outperformed them in accuracy, suggesting that for the cognitive properties we examined, a straightforward linear relationship suffices. This raises intriguing questions about the brain's representational space: is cognition more linear than previously assumed, or might nonlinear dynamics play a greater role in other cognitive domains? Recent theoretical work supports the notion that macroscale brain dynamics Nozari et al. [2024], Ivanova et al. [2022] within high-dimensional representational spaces may indeed be more linear than nonlinear, strongly corroborating our results.

Finally, we acknowledge limitations in the current study that should be addressed in future research. First, our assessment of cognitive properties, specifically emotional valence, relied on a pre-trained model. It would be valuable to consider how human ratings of the reconstructed images might influence these results. Second, our perturbation strategy was implemented solely on VDVAE and based on methodology from previous studies Goetschalckx et al. [2019]. While similar cognitive perturbation techniques have been applied to various other architectures, such as GANs and diffusion models Goetschalckx et al. [2021], it is important to test this approach within the context of human-machine alignment as discussed in this paper to establish the generalizability of our findings. Lastly, while we observed a preference for linear mappings in our analysis, our choice of nonlinear integration may have been limiting. Exploring higher-order mappings, such as those based on Taylor expansions, or implementing nonlinear mappings through neural networks could yield further insights into the nature of cognitive alignment explored in this study.

## Acknowledgments and Disclosure of Funding

This work was funded in part by von Behring-Röntgen Stiftung (Grant No. 70-00038 to HJ). The funder had no involvement in the study design, data collection, management, analysis, interpretation, report writing, or the decision to submit the report for publication. The Authors declare no competing interests.

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
