# OpenReview forum: "On the cognitive alignment between humans and machines"
_NeurIPS.cc/2024/Workshop/UniReps — UniReps_

### Official Review · Reviewer_PaNm · 2024-09-28
**Novel and promising analysis, unclear evaluation**

**Rating:** 5
**Confidence:** 2

**Review:**

Summary:
The paper examines whether the latent space representations of DNN are neurally and psychologically aligned with human brain representations, particularly investigating linear and nonlinear mapping and how interventions produce corresponding effects in different representations.

Pro:
- **Novelty**
- **Clean exposure** and accessible neuroscientific language also for a computer scientist.

Cons:
- **Representation vs. Reconstruction**: I am unclear on how the authors claim that the latent space can retrieve the cognitive properties by comparing the emotional valence score between the input and output images. Wouldn't an identity mapping return a perfect score as well? I am not a neuroscientist and probably missing the core of your experiments, but I am currently lost in this dilemma.
- **Reproducibility**: The only information provided regarding the used VAE is that the authors considered only 31 layers out of 75. Code not available (?). It is not clear how to reproduce the experiments. An Experiment Settings Section in the appendix would be appreciated (particularly with models' details).
- **Generative AI is more than VDVAE**: The authors conclude that "these findings emphasize the utility of linear mappings between brain activity and *generative models* in understanding cognition." However, they 'only' run experiments on VDVAE. Despite its valuable results, I believe more experiments with different architectures are needed to support this claim.

Typos:
- Line 15, "(r 0.4)" without context and equal sign doesn't mean anything. I assumed the authors refer to Pearson correlation, but I have understood that only reading the full paper.
- Figure 1: Mispel "fMIR" for "fMRI" twice
- Line 88: Missing References

---

### Official Review · Reviewer_xizw · 2024-10-05
**Aligning Cognitive Properties Between VDVAE Latent Spaces and fMRI Brain Responses: Exploring Emotional Valence in AI and Human Systems**

**Rating:** 7
**Confidence:** 3

**Review:**

This paper explores a novel and fascinating domain by aligning cognitive properties, particularly emotional valence, between the latent space representations of a VDVAE and fMRI brain responses in humans. The work is ambitious, addressing critical questions in cognitive neuroscience and neuro-engineering, such as whether latent spaces in AI models can reflect psychological constructs like emotional valence, and whether this alignment can be perturbed and observed across human and machine systems.

Strengths:

Innovative Approach: The paper introduces an intriguing intersection between machine learning and cognitive neuroscience. By leveraging generative models like VDVAE and fMRI data, the study breaks new ground in exploring the cognitive alignment between humans and AI models. The choice to focus on emotional valence, a well-defined and important cognitive property, is highly relevant, especially for potential clinical applications in mood disorders like Major Depressive Disorder.

Methodological Rigor: The use of the Natural Scenes Dataset, with its high-resolution 7T fMRI data and extensive set of visual stimuli, adds depth and robustness to the study. The authors thoroughly test both linear and nonlinear mappings, using ridge regression and Taylor expansion, to examine the alignment between VDVAE and brain responses.

Cross-Subject Alignment: The paper presents an important finding: aligning neural data across subjects significantly improves cognitive alignment with the VDVAE model, restoring accuracy to levels seen with machine-only data. This suggests that inter-subject variability in neural responses can be addressed, an exciting prospect for group studies in neuroscience and AI-driven cognitive interventions.

Clinical Implications: The paper's findings on cognitive perturbation and alignment have promising implications for studying mental disorders, especially with regards to individual differences in emotional biases. The potential to use AI systems to decode and modulate cognitive states could revolutionize neuro-engineering and offer new treatments for disorders like depression.

Areas for Improvement:

Nonlinear Modeling: The study’s attempt to use Taylor expansion for nonlinear mapping is notable, but the results indicate that nonlinear methods do not outperform linear ones. However, this section could benefit from a deeper discussion on why nonlinear approaches failed to yield better results. Are there specific types of nonlinearity not captured by Taylor expansions?

Broader Cognitive Properties: The paper focuses primarily on emotional valence, which is indeed a central property of cognitive processing, particularly in clinical applications. However, it would be useful to expand this exploration to other cognitive dimensions such as arousal, attention, or control to assess whether the observed human-machine alignment holds across broader aspects of cognition. This would lend further generalizability to the findings.

VDVAE Model Choice: Although the use of VDVAE is justified, it would be useful to include a brief discussion on how different types of generative models (e.g., GANs or other types of VAEs) might compare in terms of cognitive alignment. A comparison with simpler or more interpretable models might offer insights into which model architectures are most appropriate for future studies involving human brain data.

Missing figure ref l88

---

### Official Review · Reviewer_gbJz · 2024-10-07
**Review of 'On the cognitive alignment between humans and machines'**

**Rating:** 6
**Confidence:** 4

**Review:**

The paper investigates the alignment between human brain representations and the latent space of a generative computer vision model, with a focus on emotional valence. Using fMRI data of subjects exposed to visual stimuli from the Natural Scenes Dataset (NSD), the authors perform experiments to:
- Determine whether linearly mapping between a VDVAE model's latent space and fMRI data can capture emotional valence.
- Assess if perturbing the relevant dimensions produces corresponding effects between the VDVAE and human subjects.
- Compare linear and nonlinear mappings.

Key findings:
- The VDVAE latent space aligns well with fMRI responses across subjects.
- Perturbations in the relevant dimensions produce consistent effects across the VDVAE and the human subjects.
- Linear mappings perform as well as or better than nonlinear Taylor expansions.
- When the fMRI data is aligned across subjects, cognitive alignment becomes very good (r~0.7).

Strengths:
- Novelty: the first work that I’m aware of which assesses the effects of perturbing the relevant dimensions to the human - AI alignment in emotional valence. The positive results provide stronger evidence of alignment compared to just studying the [non]linear mappings using correlations.
- Interesting findings on linear vs. nonlinear mappings and cross-subject alignment.
- Reasonable methodology.
- Interesting discussion of potential implications for cognitive neuroscience and mental health research.

Weaknesses (roughly, in order of perceived importance):
- The emotional valence labels are AI-generated (not human-generated). It might be good to specify this more visibly (e.g. in the abstract), and perhaps to discuss (in more details) how this might weaken the results/claims of the paper.
- The effect sizes of the perturbations seem relatively small. The paper might benefit from discussing this more and perhaps specifying this more visibly (e.g. in the abstract).
- This work could benefit from appendices with some missing details, on e.g. how the alignment of the fMRI data of multiple subjects was performed, or on the results using nonlinear alignments between the VDVAE and fMRI data
- Some potential missing references, e.g. two previous, related papers which study the alignment between human and AI representations of emotional valence on visual stimuli: ‘The Perceptual Primacy of Feeling: Affectless machine vision models robustly predict human visual arousal, valence, and aesthetics’; ‘Controlled assessment of CLIP-style language-aligned vision models in prediction of brain & behavioral data’.
- The graphics of figure 2 seem suboptimal.
- Missing reference at line 88, page 3.

---

### Decision · Program_Chairs · 2024-10-10

**Decision:**

Accept

**Comment:**

In light of the positive reviewers' feedback and relevancy of the submission, we are pleased to accept this paper for presentation at UniReps 2024. We kindly ask the authors to incorporate the reviewers' suggestions and feedback in the final camera-ready version of the manuscript.

---

> ### Author Response · Authors · 2024-11-06
> **Camera Ready**
>
> We extend our sincere gratitude to the reviewers and program chairs for their valuable and constructive feedback on our manuscript. In this revised, camera-ready version, we have addressed these comments by expanding the discussion on our work’s limitations, including missing citations, correcting typographical errors, and clarifying our methodology with additional details in Section 2.4. We are excited by the acceptance of our paper and look forward to engaging in more discussions during the workshop.